# Vitamin D Levels in Pregnant Women Do Not Affect Neonatal Bone Strength

**DOI:** 10.3390/children9060883

**Published:** 2022-06-13

**Authors:** Orly Levkovitz, Elena Lagerev, Sofia Bauer-Rusak, Ita Litmanovitz, Eynit Grinblatt, Gisela Laura Sirota, Shachar Shalit, Shmuel Arnon

**Affiliations:** 1Department of Neonatology, Meir Medical Center, Kfar Saba 4428163, Israel; bauers@clalit.org.il (S.B.-R.); italitman@gmail.com (I.L.); sheynitgr@clalit.org.il (E.G.); gisela.sirota@clalit.org.il (G.L.S.); shachar.shalit@clalit.org.il (S.S.); shmuelar@clalit.org.il (S.A.); 2Department of Neonatology, Assuta-Ashdod Medical Center, Ashdod 7747629, Israel; elenalag@assuta.co.il; 3Sackler School of Medicine, Tel Aviv University, Tel Aviv 6997801, Israel

**Keywords:** neonate, pregnancy, quantitative ultrasound, speed of sound, 25-hydroxy vitamin D, bone strength, nutrition

## Abstract

Vitamin D plays a key role in regulating calcium and phosphate metabolism. However, whether maternal vitamin D levels affect fetal bone strength is unclear. This study assessed correlations between maternal 25(OH)D status and neonatal bone strength 25(OH)D levels, these were measured in the maternal and infant cord blood of 81 mother–infant dyads. Bone strength was measured using a quantitative ultrasound (QUS) of tibial bone speed of sound (SOS). Maternal vitamin D intake, medical history and lifestyle were evaluated from questionnaires. Maternal 25(OH)D levels were deficient (<25 nmol/L) in 24.7%, insufficient (25–50 nmol/L) in 37% and sufficient (>50 nmol/L) in 38.3%. The maternal and cord blood 25(OH)D levels correlated (r = 0.85, *p* < 0.001). Cord blood levels (57.9 ± 33.5 nmol/L) were higher than the maternal blood levels (46.3 ± 23.2: *p* < 0.001). The mean SOS was 3042 ± 130 m/s. The neonatal SOS and 25(OH)D levels were not correlated. The mean bone SOS levels were comparable in the three maternal and cord blood 25(OH)D groups. No correlation was found between the maternal 25(OH)D levels and the neonatal anthropometrics. Although the 25(OH)D levels were higher in Jewish mothers than they were in Muslim mothers (51.1 ± 22.6 nmol/L vs. 24 ± 14.7 nmol/L, respectively: *p* = 0.002) and in those who took supplemental vitamin D, the bone SOS levels were comparable. In conclusion, maternal vitamin D levels correlate with cord levels but do not affect bone strength or growth parameters.

## 1. Introduction

Vitamin D plays a key role in regulating calcium and phosphate metabolism. It increases the absorption of calcium and phosphate through the intestines and their reabsorption through the renal tubules. It also provides the minerals needed to form new bone. The complications of vitamin D deficiency can include secondary hyperparathyroidism, bone resorption and the under-mineralization of new bone, leading to osteomalacia and rickets [1]. Major risk-factors for vitamin D deficiency include maternal skin pigmentation, the use of sun protective clothing, nutrition and dietary supplements [2].

During pregnancy, the mother’s physiology adapts to provide the needed calcium and phosphate to both the developing fetus and to herself, mainly by doubling the intestinal absorption of these minerals [3]. Minerals transported across the placenta are used to form the fetal skeleton. Maternal 25-hydroxy vitamin D (25(OH)D) and 1,25 hydroxy vitamin D, or calcitriol (the active form of vitamin D) readily cross the placenta to supply the fetus [4]. Maternal 25(OH)D is transformed to calcitriol by the maternal and fetal kidneys, the placenta and, probably, by the extrarenal maternal and fetal tissues [5].

The fetal skeletal growth and mineralization has a complicated physiological mechanism. The effect of vitamin D on the development and growth of the fetal skeleton is poorly understood, and the scientific literature is inconsistent. Studies on animal models indicate that calcitriol and vitamin D receptors are not required for normal adaptation during pregnancy [6,7,8,9], a mechanism provided to ensure mineral delivery to the developing fetus, independent of vitamin intake or synthesis, seasonal variations or sun exposure. Human data are contradictory. Epidemiological studies have reported that vitamin D deficiency does not usually lead to hypocalcemia or rickets until weeks or months after birth, with the incidence peaking at 6 to 18 months [4,10]. A trial comparing infants to mothers who were supplemented with high-dose versus standard-dose vitamin D, found that, despite higher maternal 25(OH)D levels in the treated group, there were no benefits to the neonate in the first 10–14 days postpartum, in anthropomorphics or body composition [11]. On the other hand, studies investigating bone variables in neonates reported that infants born to mothers with higher 25(OH)D blood concentrations had higher bone mineral concentrations and cross-sectional surface area [12]. Alterations in distal femoral morphology and reduced volume in utero were associated with mild maternal vitamin D deficiency, based on ultrasonographic measurements [13,14]. There might be an epigenetic relation between maternal 25(OH)D levels and the accumulation of fetal bone mass [15].

One way to assess bone strength is by using a quantitative ultrasound (QUS) measurement of the bone speed of sound (SOS). This methodology is cost-effective, easy to administer and does not include ionizing radiation. QUS SOS measures the axial transmission of ultrasound waves through the bone, based on their ability to pass through bone more rapidly than through soft tissue. Unlike measuring bone mineral density and content, QUS also assesses qualitative parameters, including cortical thickness, bone elasticity, and damage from fatigue, that allow a better estimation of bone strength [16]. Previous studies suggested that measurements of bone SOS can be used to evaluate bone strength in term and premature neonates [17,18,19,20].

The primary aim of this study was to examine the relationship between maternal 25(OH)D levels and their offspring’s bone strength in a cohort of Israeli–Jewish and Muslim mother–infant pairs, using QUS SOS. We hypothesized that infants of vitamin D deficient mothers would have decreased bone strength. The secondary aim was to assess the relationship between the maternal and neonatal 25(OH)D levels, and their relationship to maternal variables associated with vitamin D deficiency.

## 2. Materials and Methods

The study was based on a community-based cohort of mother–offspring dyads. Healthy pregnant women with an uneventful pregnancy (37–42 weeks), who were admitted to the delivery room at the Meir Medical Center, Kfar-Saba, Israel, during a 20-month period, were offered enrollment on the study. Mothers with a family history of bone disease, or pregnancy complicated by diabetes, IUGR, congenital infection or congenital anomaly were excluded. The study protocol was approved by the medical center’s ethics committee. All parents provided written informed consent in accordance with the declaration of Helsinki (approval no. 0014-12-MMC2).

**Data collection:** At the time of recruitment, participants received an extensive questionnaire, including medical history, sun exposure (less than 15 min or more than 30 min per day), clothing habits (covering arms, legs, neither or all), use of sunscreen and their dietary habits. They also completed a food frequency questionnaire (FFQ) and information on the use of supplements. The FFQ was used to calculate vitamin D intake. All questionnaires were evaluated by one researcher (SS). Pregnancy follow-up and delivery records, including gestational age, birth weight and length were obtained from electronic medical data.

**Measurement of 25(OH)D concentration:** Maternal blood samples were collected in the delivery room. Venous samples of cord blood, representing neonatal levels, were obtained immediately after delivery. Blood samples were centrifuged within 2 h and stored at −30 °C until analyzed. Serum 25(OH)D concentrations were measured by a chemiluminescence assay, using a Liason instrument with the Diasorin kit (Diasorin Inc., Stillwater, MN, USA). Samples from the mother and neonatal dyads were measured simultaneously, to avoid inter-assay variations.

**QUS Measurements of Bone SOS:** A bone ultrasound was conducted during the first two days of life. SOS in the left tibia was measured using QUS (Sunlight Premier Software, Omnisense 000/8000, BeamMed Ltd., Petah Tikva, Israel). This method is designed to measure SOS at several skeletal sites, based on axial transmission. All SOS measurements were conducted by the same investigator (YL), following a standard protocol. The probe was placed in the middle of the tibia, which is halfway between the apex of the medial malleolus and the distal patellar apex. After the machine was calibrated using a standard phantom, the mean value of three measurements that were taken from the same point was calculated. The results were expressed in meters per second. Measurement accuracy was 0.25–0.5%, with a root mean square coefficient of variation of 0.4–0.8%.

**Statistical analysis:** The data are presented as mean ± SD for continuous variables and as numbers and percentages for nominal parameters. Differences between the groups were analyzed using chi-square for non-metric variables and the Mann-Whitney non-parametric test for metric data, which were not normally distributed (Shapiro-Wilk test). The Pearson correlation was calculated to determine the relationship between the different parameters and the maternal 25(OH)D levels. *p* < 0.05 was considered statistically significant. The statistical analysis was performed using SPSS-28 software (IBM, Armonk, NY, USA).

## 3. Results

Eighty-one mother–infant pairs were enrolled. Their characteristics are described in Table 1. Maternal and umbilical blood samples were successfully drawn from 79 mother–infant pairs for 25(OH)D analysis. The mean maternal 25(OH)D level was 46.3 ± 23.2 nmol/L. They were deficient (<25 nmol/L) in 24.7%, insufficient (25–50 nmol/L) in 37% and sufficient (>50 nmol/L) in 38.3%. The mean cord blood 25(OH)D level (representing neonatal levels) was 57.9 ± 33.5 nmol/L. The cord blood 25(OH)D levels were deficient in 12.3%, insufficient in 40.7% and sufficient in 47%. The maternal and cord blood 25(OH)D levels were positively correlated (r = 0.85, *p* < 0.001: Figure 1). A significant correlation between the 25(OH)D levels in the mothers’ and the infants’ cord blood was also found in each category (*p* < 0.001). The mean cord blood levels (57.9 ± 33.5 nmol/L) were higher in comparison with the maternal blood levels (46.3 ± 23.2 nmol/L: *p* < 0.001).

The left tibial SOS measurement was successfully obtained from 74 neonates, with a mean value of 3042 ± 130 m/s. No correlation was found between the SOS measurements and the maternal or cord blood 25(OH)D levels. Accordingly, the mean SOS levels were comparable among the three 25(OH)D groups of maternal and cord blood (Table 2).

No significant correlation was noted between the maternal 25(OH)D levels and the neonatal birth weight (*p* = 0.47), length (*p* = 0.94) or head circumference (*p* = 0.063).

In terms of ethnicity, the mean 25(OH)D level of Jewish mothers was significantly higher, compared to Muslim–Arab mothers (51.1 ± 22.6 nmol/L vs. 24 ± 14.7 nmol/L, respectively: *p* = 0.002). A similar finding was also observed in the cord blood of the two ethnic groups (62.4 ± 33.6 nmol/L vs. 24.7 ± 14.2 nmol/L, respectively: *p* = 0.001). The SOS measurements were comparable (3046 ± 128 vs. 2974 ± 122 m/s, respectively: *p* = 0.13). Maternal 25(OH)D levels were not correlated with maternal age, sunlight exposure or sunscreen use. No seasonal variations were found (Table 3). Vitamin D supplementation during pregnancy was reported by 45% of the mothers. The maternal 25(OH)D levels correlated with the use of vitamin D supplementation, but not with vitamin D consumption from food.

## 4. Discussion

Neonatal bone measurements can reflect the status of bone development in utero and provide important information for understanding bone metabolism and the possible development of osteoporosis in the future. The results presented here indicate a high incidence of suboptimal (deficient and insufficient) 25(OH)D levels in a cohort of pregnant Israeli women. Maternal and cord blood 25(OH)D levels were strongly correlated, but the 25(OH)D levels and neonatal bone strength were not, as reflected by the QUS SOS measurements.

The prevalence of deficient and insufficient 25(OH)D levels (less than 50 nmol/L) in our study was 61.7% in maternal blood, and 53% in cord blood. Numerous studies regarding vitamin D deficiency across different populations worldwide reported great variability–from less than 20% to above 80% [2]. A previous study from Israel demonstrated maternal 25(OH)D levels below 25 nmol/L in approximately 30% of the pregnant women, with significant differences even across different cultural groups within the country [21]. Our study found a positive correlation between the 25(OH)D levels in cord blood and in maternal blood. Although most studies reported lower 25(OH)D levels in neonates’ cord blood in comparison to maternal 25(OH)D levels [4,22,23,24,25], we found higher levels of 25(OH)D in neonates compared to their mothers. A similar phenomenon was also observed by others [26,27]. Esmeraldo et al. found that mean 25(OH)D concentrations of 48.7 ± 15.2 ng/mL vs. 26.0 ± 6.7 ng/d in newborns and their mothers, respectively. In 92% of newborns, 25(OH)D levels were sufficient, but this was only the case in 25.8% of the mothers [27]. The exact mechanism and the factors that affect the 25(OH)D transfer across the placenta are still elusive. The metabolism of 25(OH)D by the placenta and the fetal kidneys, and the relationship between 25(OH)D and the vitamin D binding protein, may have a role in the vitamin D kinetics and levels measured in both the pregnant mother and the fetus [28,29,30].

Despite the correlation between the maternal blood and cord blood 25(OH)D levels, we did not find a correlation between maternal 25(OH)D levels and neonatal bone strength, as reflected by the QUS SOS measurements. Other studies have addressed the correlation between these two variables, with conflicting findings. In agreement with our results, Velkavrh et al. [31] did not find a significant association between maternal 25(OH)D during pregnancy and infants’ bone SOS in 73 pairs of mothers and their term neonates. Liao et al. [32] found a positive correlation between maternal 25(OH)D and bone SOS in Chinese mother–infant pairs. This correlation, although statistically significant, was weak (r = 0.399) and virtually all mothers in Liao’s study were severely 25(OH)D deficient. A recent review of observational, cohort and randomized controlled trials [33] showed conflicting results regarding the effect of maternal vitamin D on their offspring’s bone density and other bone measurements, both in observational and interventional studies. In contrast to the conflicting evidence of the effects of maternal vitamin D status on fetal and neonatal bone, the literature is more consistent regarding the association between maternal vitamin D status during pregnancy and their children’s bone parameters later in childhood. Hyde et al. [34] found a positive correlation between maternal 25(OH)D in early pregnancy and bone SOS at the age of 11 years. Javaid et al. found that maternal vitamin D insufficiency during pregnancy was associated with reduced bone mineral accrual in the offspring at the age of 9 years [35]. Other studies demonstrated an association between bone parameters in childhood, and variables that may be related to maternal vitamin D status, such as UVB exposure [36] and nutrition [37] during pregnancy. It was suggested that mechanisms other than vitamin D are responsible for intestinal calcium absorption in fetal life, and that the role of vitamin D in bone metabolism becomes more significant in postnatal life [38].

We did not find a correlation between maternal 25(OH)D and infant birth weight, length and head circumference. Our results agree with those of several previous reports [31,35,39,40], whereas other studies found a positive association between maternal 25(OH)D and birth weight [41,42,43]. The results of a recent Cochrane review [44] and previous meta-analyses of randomized controlled trials [45,46,47] were inconclusive regarding the relationship between vitamin D and birth weight.

Our study revealed lower 25(OH)D levels in mothers and offspring of Muslim–Arab origin, compared to those of Jewish origin. These findings agree with previous studies, which reported a high prevalence of 25(OH)D deficiency in Arab populations [22,25,48,49]. We found no significant differences in the nutritional habits, sun exposure or clothing style between the two populations, that could explain the differences in 25(OH)D levels, as suggested in other studies [25,50]. Other differences in neonatal morbidities were reported between the Jewish and Arab populations in Israel [51], suggesting that genetic and epigenetic factors may contribute to these differences.

As previously reported, we found a positive correlation between vitamin D supplementation and 25(OH)D levels in the mothers and infants. These results are in agreement with an interventional study that demonstrated a more significant effect of supplements on the 25(OH)D levels of pregnant women, compared to sunlight exposure [52]. However, a review of randomized interventions with vitamin D supplementation of pregnant women revealed no changes in cord blood calcium, phosphate, PTH, birth weight or anthropometric measurements in babies of vitamin D-supplemented mothers, compared to babies of placebo or low-dose-treated mothers [38].

To our knowledge, only a few studies have addressed the relationship between maternal 25(OH)D levels and their offspring’s bone strength, as reflected by QUS SOS, and the results have been inconsistent. The current study provides additional information regarding this topic, including ethnic influences on neonatal outcomes in relation to vitamin D. QUS is an inexpensive, radiation-free tool that is available to use at the bedside. It provides information on bone composition and bone function. Several studies have shown comparable results between QUS and dual-energy X-ray Absorptiometry (DXA), which is considered the gold standard, in various clinical situations. However, other research studies did not find an agreement between the two methods. Therefore, the use of QUS to assess bone health is still controversial [53,54,55].

There were some limitations to this study. Although this is one of the largest studies of the association between 25(OH)D levels and neonatal bone strength, the small subgroup sizes, primarily of vitamin D-deficient subjects (6 neonates and 12 mothers), may influence our ability to detect changes and trends in bone strength in this group. We did not follow the 25(OH)D levels at different periods during pregnancy, which could have provided additional information on potentially critical periods during the early stages of pregnancy and their effect on neonatal bone strength.

## 5. Conclusions

The current study demonstrates that maternal 25(OH)D levels correlate with cord levels but do not affect bone strength or growth parameters. Higher levels of 25(OH)D were found in cord blood, compared to maternal blood. Additional studies are needed to provide more information on the effects of vitamin D on fetal development.

## Figures and Tables

**Figure 1 children-09-00883-f001:**
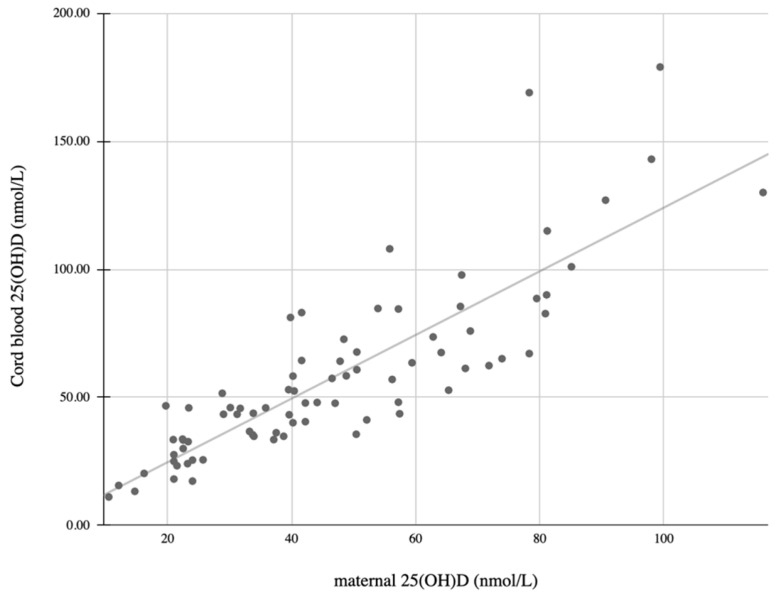
Linear correlation between maternal and cord blood 25(OH)D levels (r = 0.85, *p* < 0.001).

**Table 1 children-09-00883-t001:** Patients’ characteristics.

Maternal	Value
Age (years)	30.7 ± 5.1
Education (years)	15.6 ± 2.2
Origin: Jewish	69 (86%)
Nulliparous	47 (59%)
Singleton	75 (94%)
Daily sun exposure	
Less than 15 min	31 (40%)
More than 30 min	47 (60%)
Sun skin protection	29 (37%)
Vitamin D supplementation	36 (45%)
Vitamin D supplementation dose (IU)	1127 ± 894
Vitamin D from nutrition (IU)	239.3 ± 140.6
Total Vitamin D consumption (IU)	748 ± 824
**Infant**	
Gestational age (weeks)	39.5 ± 1.26
Birth weight (grams)	3223 ± 476
Length (centimeters)	50.7 ± 2.9
Head circumference (cm)	34.3 ± 1.3
Male sex	35 (44%)
Speed of sound (m/s)	3042 ± 130

IU—international units.

**Table 2 children-09-00883-t002:** Speed of sound among infants, according to 25(OH)D level.

25(OH)D (nmol/L)	N	SOS (m/s) (Mean ± SD)	*p*-Value
**Cord blood**			
<25	6	3040 ± 161	0.97
25–50	29	3049 ± 137	
>50	38	3042 ± 123	
**Maternal**			
<25	13	3053 ± 119	0.15
25–50	29	3076 ± 143	
>50	31	3011 ± 118	

SOS—Speed of sound.

**Table 3 children-09-00883-t003:** Correlation between maternal 25(OH)D and maternal and infant variables.

**Continuous Variables**	**Pearson Correlation**	**Significance**
Infant 25(OH)D levels	0.85	<0.001
Infant SOS	−0.1	0.4
Vitamin D intake–total	0.36	0.001
Vitamin D intake from food only	0.026	0.82
Vitamin D intake from supplements	0.21	0.002
Infant length at birth	0.009	0.94
Infant weight at birth	0.082	0.47
Infant head circumference	0.212	0.063
Maternal age	0.11	0.32
**Non-continuous variables (MW test)**		**Significance**
Maternal ethnicity		<0.0001
Maternal sunlight exposure		0.62
Maternal skin exposure		0.12
Maternal use of sunscreen		0.08
Season of the year		0.76

## Data Availability

The data presented in this study are available on request from the corresponding author.

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
