# Peer review of "Vitamin D Levels in Pregnant Women Do Not Affect Neonatal Bone Strength"

_children, 2022, doi:10.3390/children9060883_

Round 1

Reviewer 1 Report

This is an intriguing manuscript which reports the use of the novel technique of quantitative ultrasound (QUS) measurements through the tibia of newborn infants to deduce the bone strength and then to relate this to the serum 25(OH)D concentrations in cord blood and in maternal blood. Several points to consider:

  1. Most studies of bone, related to vitamin D status, have estimated bone density using dual-energy x-ray absorptiometry (DEXA). It would be of assistance to the reader if some information were provided to compare the use of QUS and DEXA in assessing the normality or abnormality of bone. Is it possible, for example, that although the QUS values showed no relationship with the serum 25(OH)D concentrations, that there could still be differences in bone density of newborn infants as measured by DEXA?
  2. At several points throughout the manuscript the term “vitamin D level” is used when what was actually measured was 25(OH)D. In many serum samples both vitamin D and 25(OH)D are present. Therefore, to avoid confusion, the concentrations being measured should be referred to as “25(OH)D level”. This correction is needed at lines: 59, 63, 240, 248-249, 260, 264 and in the first line of data in Table 3.
  3. Lines 76-77: “ bone SOS measurements can be used for to evaluate bone strength” should read: “bone SOS measurements can be used to evaluate bone strength”
  4. In Table 1 the data row for “Vitamin D supplementation dose 1127±894” needs a definition of the units for the numbers being quoted.

Reviewer 2 Report

Introduction

The fragment between lines 69 – 77 should be in the Discussion, not in the Introduction.

Discussion

The results obtained in this study confirm the lack of a positive correlation between the concentration of 25(OH)D in the maternal blood and the infant’s bone strength or growth parameters. They also reveal differences in calcidiol levels, contrary to expectations, such as a higher level in umbilical cord blood compared to the mother's blood, or much milder deficiencies in fetuses than in deficient mothers. Other authors have also obtained such results and, indeed, they are not easy to discuss and explain the mechanism.

1.       In this work, the authors should use the cited literature more carefully, not make suggestions that are already proven facts (eg lines 225-226). For example, concerning  the observed lack of expected correlations in vitamin D levels, and neonatal anthropometrics the authors suggest/guess that the effect of vitamin D on the developing bone is more prominent later in life, during the postnatal period. Meanwhile, this is not a suggestion, but confirmed observations from studies by other authors, cited in this paper, such as Hyde et al. [33] or Javaid et al. [35] but not only by them. The long-term consequences of prenatal vitamin deficiency noted also Sayers A. (2009), J Clin Endocrinol Metab, or Cole et al., J Bone Miner Res (2009).

2.       The authors summarize the results obtained, but do not seek an explanation based on other works. They emphasize, however, that the work introduces a new look at the influence of the ethnic factor on the neonatal outcome. However, the relevant fragment of the Discussion (lines between 233-245) is not an analysis of the results in this context, but is limited to one general sentence.

3.       Maybe some elements of originality should be emphasized - e.g. the bone strength test method and the results should be discussed in the context of the marked parameters.

4.       The authors reverse the importance of certain phenomena, considering their results to be supported by the majority of the authors. For example, they write that, in contrast to the results obtained in this study, some authors noted lower 25(OH)D levels in umbilical cord blood compared to maternal blood and they imply that this was an intermittent effect (lines between 189 – 192). Whereas, the usual level of 25 (OH) D in umbilical cord blood is lower, accounting for about 70-100% of that in maternal blood. Consequently, the offspring of deficient mothers will also be deficient. However, such dependencies are somewhat opposed to the results obtained in this study.

5.       Lines between 216-219: discussing references No.: [28]; [30] and [34], the authors do not comment on the difference between the maternal and fetal calcitriol levels, expressing some surprise with the fact that the fetal calcitriol level does not positively correlate with the maternal 1,25(OH)2D level (increased by up to 100% during pregnancy), despite such a relationship in the case of 25(OH)D. It is  suggested to read the work by Christesen et al. (ACTA Obstet Gyneol Scand, 2012; 91), which explains the known mechanism of this phenomenon.

6.       Lines between 249-251: too one-sided statement. It is suggested to get acquainted with the following articles: Christesen et al. (as above), Lapillone A (Med Hypotheses, 2010) and Dror DK (Curr Opin Obstet Gynecol, 2011) - the results presented in these studies contradict this statement.

Round 2

Reviewer 2 Report

Page 9/13 -  the abbreviation DXA should be explained.